# Seroprevalence of SARS-CoV-2 antibodies and retrospective mortality in two African settings: Lubumbashi, Democratic Republic of the Congo and Abidjan, Côte d'Ivoire

Erica Simons[1], Birgit Nikolay[1]*, Pascal Ouedraogo[1], Estelle Pasquier[1], Carlos Tiemeni[2], Ismael Adjaho[3], Colette Badjo[3], Kaouther Chamman[1], Mariam Diomandé[3], Mireille Dosso[4], Moussa Doumbia[4], Yves Asuni Izia[2], Hugues Kakompe[5], Anne Marie Katsomya[2], Vicky Kij[5], Viviane Kouakou Akissi[4], Christopher Mambula[2], Placide Mbala-Kingebeni[6], Jacques Muzinga[7], Basile Ngoy[5], Lou Penali[4], Alessandro Pini[1], Klaudia Porten[1], Halidou Salou[1], Daouda Sevede[4], Francisco Luquero[1], Etienne Gignoux[1]

1 Epicentre, Paris, France, 2 Médecins Sans Frontières, Paris, France, 3 Médecins Sans Frontières, Abidjan, Cote d'Ivoire, 4 Institut Pasteur Cote d'Ivoire, Abidjan, Cote d'Ivoire, 5 Ministry of Health, Kinshasa, Democratic Republic of the Congo, 6 Institut National de Recherche Biomédicale, Kinshasa, Democratic Republic of the Congo, 7 Laboratoire National de Lubumbashi, Lubumbashi, Democratic Republic of the Congo

☯ These authors contributed equally to this work.

* birgit.nikolay@epicentre.msf.org

**Data Availability Statement:** The minimal dataset underlying the findings of this study is available on

## Abstract

Although seroprevalence studies have demonstrated the wide circulation of SARS-COV-2 in African countries, the impact on population health in these settings is still poorly understood. Using representative samples of the general population, we evaluated retrospective mortality and seroprevalence of anti-SARS-CoV-2 antibodies in Lubumbashi and Abidjan. The studies included retrospective mortality surveys and nested anti-SARS-CoV-2 antibody prevalence surveys. In Lubumbashi the study took place during April-May 2021 and in Abidjan the survey was implemented in two phases: July-August 2021 and October-November 2021. Crude mortality rates were stratified between pre-pandemic and pandemic periods and further investigated by age group and COVID waves. Anti-SARS-CoV-2 seroprevalence was quantified by rapid diagnostic testing (RDT) and laboratory-based testing (ELISA in Lubumbashi and ECLIA in Abidjan). In Lubumbashi, the crude mortality rate (CMR) increased from 0.08 deaths per 10 000 persons per day (pre-pandemic) to 0.20 deaths per 10 000 persons per day (pandemic period). Increases were particularly pronounced among <5 years old. In Abidjan, no overall increase was observed during the pandemic period (pre-pandemic: 0.05 deaths per 10 000 persons per day; pandemic: 0.07 deaths per 10 000 persons per day). However, an increase was observed during the third wave (0.11 deaths per 10 000 persons per day). The estimated seroprevalence in Lubumbashi was 15.7% (RDT) and 43.2% (laboratory-based). In Abidjan, the estimated seroprevalence was 17.4% (RDT) and 72.9% (laboratory-based) during the first phase of the survey and 38.8% (RDT) and 82.2% (laboratory-based) during the second phase of the survey. Although circulation of

request, in accordance with the legal framework set forth by Médecins Sans Frontières (MSF) data sharing policy (Karunakara U, PLoS Med 2013). MSF is committed to share and disseminate health data from its programs and research in an open, timely, and transparent manner in order to promote health benefits for populations while respecting ethical and legal obligations towards patients, research participants, and their communities. The MSF data sharing policy ensures that data will be available upon request to interested researchers while addressing all security, legal, and ethical concerns. All readers may contact data.sharing@msf.org or Nouha TOUATI (nouha.touati@epicentre.msf.org) to request data.

**Funding:** Médecins Sans Frontières funded this study and provides core funding to Epicentre. Médecins Sans Frontières, the funder of this study, participated in the design of the study, the collection and analysis of the data and the writing of this manuscript.

**Competing interests:** The authors have declared that no competing interests exist.

SARS-CoV-2 seems to have been extensive in both settings, the public health impact varied. The increases, particularly among the youngest age group, suggest indirect impacts of COVID and the pandemic on population health. The seroprevalence results confirmed substantial underdetection of cases through the national surveillance systems.

## Introduction

Official surveillance data from African countries suggest that the public health impact of the severe acute respiratory syndrome coronavirus 2 (SARS-CoV-2) associated with coronavirus disease (COVID-19) is less than that observed in Asia, America, and Europe [1]. Several factors, including the younger population, early government actions, lower prevalence of comorbidities, cross-immunity, environmental factors and surveillance, have been noted as possible explanations for the lower mortality observed in most African countries [2]. While seroprevalence studies have been conducted in several African countries indicating widespread circulation of the virus in contrast to often low surveillance figures, few population-based mortality studies have been conducted to measure the impact of this high virus circulation [3].

Here we studied the public health impact of SARS-CoV-2 in two African countries with low numbers of reported COVID-19 associated deaths, the Democratic Republic of Congo (DRC) and Côte d'Ivoire. The first COVID-19 cases were reported nearly simultaneously in the two countries, on 10 March 2020 in the DRC and on the following day in Côte d'Ivoire. After the first wave, a seroprevalence of 16.6% among the general population was observed in Kinshasa, the capital city of the DRC [4], and 25.1% among gold mine workers in Côte d'Ivoire [5]. Côte d'Ivoire was among the first African countries to receive COVID-19 vaccines with vaccination campaigns starting in March 2021. In the DRC, COVID vaccination campaigns began in April 2021, but were temporarily suspended shortly afterwards. In Lubumbashi, the studied area in the DRC, vaccinations began in May 2022. As of the start of the respective surveys (12 April 2021 in DRC and 15 July 2021 in Côte d'Ivoire), 28,542 cases and 745 deaths had been reported in the DRC and 48,999 cases and 319 deaths reported in Côte d'Ivoire [6].

To quantify the extent of SARS-COV-2 infections and to detect potential increases in the crude mortality rate (CMR) during the SARS-COV-2 pandemic phase, we assessed seroprevalence of anti-SARS-CoV-2 antibodies and retrospective mortality in a representative sample of the general population in three health zones of Lubumbashi, DRC and two communes of Abidjan, Côte d'Ivoire more than one year after the first confirmed cases of COVID-19 in these settings.

## Methods

We conducted two cross-sectional household-based surveys in the cities of Lubumbashi (DRC) and Abidjan (Côte d'Ivoire). The survey in Lubumbashi took place in the health zones Lubumbashi (stratum 1) and Kampemba/Tshamilemba (stratum 2), which were chosen based on COVID incidence in the initial months and increased reporting of deaths through cemetery surveillance, respectively. In Abidjan, the survey was implemented in the communes of Marcory (stratum 1) and Yopougon (stratum 2). These urban areas were chosen because, while the prevalence of poverty is higher in Yopougon than Marcory [7], Yopougon had a much lower COVID-19 attack rate than Marcory according to available surveillance data [8].

The studies each included a retrospective mortality survey using a two-stage cluster sampling methodology and a nested SARS-CoV-2 antibody prevalence survey. Briefly, 710 clusters

(Lubumbashi) and 640 clusters (Abidjan) were first randomly selected based on spatial sampling and subsequently 5 randomly selected households per cluster were included in the mortality survey and 1 household per cluster in the seroprevalence survey. A detailed description of the study design and sampling methodology is provided in S1 Text. The mortality questionnaire consisted of sections covering housing characteristics, demographic data, and information on deaths (including the date of death) that occurred during the recall period. For households included in the seroprevalence survey, individual questionnaires were administered to each household member or their parent/guardian covering socio-demographic information, medical history, potential SARS-CoV-2 exposures, history of any possible COVID-related symptoms since 2020 and COVID-19 vaccination status. Generic questionnaire tools that were slightly adapted to study settings are provided in S2 Text.

All members of households included in the seroprevalence survey were asked to provide a blood sample, either in the form of dried blood spots using finger or heel pricks (participants of all ages in Lubumbashi and children less than 5 years old in Abidjan) or in the form of venous blood (participants 5 years and older in Abidjan). In both sites, serological testing was done using a rapid serological test (BIOSYNEX® COVID-19 BSS (IgG/IgM) (Biosynex SA, Switzerland). The Biosynex test has shown good performance (sensitivity: 95.8% [95%CI 90.2 −100.0], specificity: 98.1% [95%CI 94.3–100.0])) [9]. Additional laboratory-based testing was conducted using the EUROIMMUN Anti-SARS-CoV-2 enzyme-linked immunosorbent assay (EUROIMMUN Medizinische Labor diagnostika AG, Lübeck, Germany) at the INRB laboratory in Lubumbashi or the Roche Elecsys anti-SARS-CoV-2 S immunoassay (Roche Diagnostics, Rotkreuz, Switzerland) at the Institut Pasteur Côte d'Ivoire in Abidjan. Initial laboratory evaluations of the EUROIMMUN test showed a sensitivity of 90% [95%CI 74.4–96.5] and a specificity of 100% [95%CI 95.4–100] [10]. Initial evaluation of the Roche assay found a sensitivity of 98.8% (95% CI: 98.1–99.3%) among patients for 14 days or later after diagnosis with PCR and a specificity of 100% (95% CI: 99.7–100%) and was selected in part due to its strong performance detecting antibodies even several months after infection [11,12].

Crude mortality rates (CMR, expressed as deaths/10,000 people/day) and 95% confidence intervals (95%CI) were calculated taking into account the study design using the survey package in R. The analysis was stratified between pre-pandemic and pandemic time periods. In Lubumbashi the pre-pandemic period was defined as 1 January 2020–12 April 2020 and the pandemic period as 13 April 2020 –date of survey. In Abidjan, the pre-pandemic period was defined as 1 January 2019–12 April 2020 and the pandemic period as 13 April 2020 –date of survey. 12 April 2020 was chosen as cut-off date between the pre-pandemic and pandemic period as it was the Easter Sunday, a date easy to recall for survey participants. Differences in mortality rates were further investigated by time periods corresponding approximately to individual COVID-19 waves (Lubumbashi Wave 1: 13 April 2020–31 August 2020 and Wave 2: 1 November 2020 –date of survey; Abidjan Wave 1: 13 April 2020 –& August 2020, Wave 2: 1 January 2021–30 June 2021 and Wave 3: 1 July 2021 –date of survey). Deaths were attributed to the time periods based on their date of death. To quantify differences of the mortality rates between the pre-pandemic and pandemic periods, we estimated rate ratios with 95%CIs using a Poisson generalized linear model (GLM) with log-transformed follow-up time as offset in the survey package in R and tested for statistical significance of differences using Wald-test.

For the estimation of seroprevalence, a positive rapid diagnostic test (RDT) result was defined as positive IgM, positive IgG or positive IgM and IgG. A positive ELISA/ECLIA result was defined using the manufacturer-specified cut-off value (Euroimmun ELISA: optical density ratio $\geq 1.1$; Roche ECLIA: titer $\geq 0.8$ U/mL). Seroprevalence and 95% confidence intervals (95% CI) were estimated using the survey package in R, weighting for demographic differences between the survey sample and the general population and adjusting for the design effect. In

Abidjan, seroprevalence estimates excluded those who self-reported already having received at least one COVID-19 vaccine dose. Adults 18 years and older were eligible for vaccination during study implementation. Vaccines became available in Lubumbashi after the completion of the study. To compare seroprevalence among sub-groups (sex, age groups, strata), we estimated odds ratios (OR) and 95%CIs and tested for statistical differences using Wald test in a logistic regression model considering the study design using the survey package in R.

The proportion of symptomatic and proportion of vaccinated individuals, together with 95%CIs, were estimated using the same statistical procedure as for seroprevalence estimates. Differences in other characteristics between time periods (symptoms, comorbidities–measured in proportions) were evaluated based on Fisher's exact test.

We further investigated socioeconomic risk factors (household type, presence of latrines, and number of people per room) for a household reporting a death using a GLM logistic regression model at the household level. To investigate if seropositivity was associated with the presence of another seropositive household member, we implemented a GLM logistic regression model at individual level using the survey package taking the survey design into account.

### Ethics statement

The study in Lubumbashi has been approved by the Comité d'Etique Medicale of the University of Lubumbashi (ID UNILU/CEM/020/2020) and the study in Abidjan by the Comité National d'Etique des Sciences de la Vie et de la Santé (ID 054-21/MSHP/CNESVS-km). Both studies have been approved by the MSF ERB (ID 2089b, 2089d). Formal written consent was obtained from all participants 18 years or older. For participants less than 18 years, written consent was obtained from the parent/guardian; additional assent was obtained from participants aged 8–17 years in Lubumbashi and 9–17 years in Abidjan.

## Results

### Study population

In Lubumbashi, the mortality and seroprevalence surveys took place concurrently 12 April– 18 May 2021, at the end of the second SARS-CoV-2 wave (Fig 1A). A total of 3,506 households, including 19,694 household members, participated in the mortality survey. The median age of participants was 18 years (interquartile range [IQR] 7–32) and 49.6% were male. A total of 2,038 individuals from 650 households participated in the RDT-based serosurvey. Participants had a median age of 22 years (IQR 10–36) and 44.6% were male. Both diagnostic type results, RDT and ELISA, were available for 1,897 individuals.

In Abidjan, the surveys took place during two phases due to extenuating circumstances during data collection: 15 July– 14 August 2021 (after the second wave and during the beginning of the third wave) & 20 October– 10 November 2021 (end of and after the third wave) (Fig 1B). A total of 3,180 households, including 15,454 household members, participated in the mortality survey. The median age of participants was 25 years (IQR 13–39) and 44.3% were male. A total of 1,862 individuals from 634 households participated in the RDT-based serosurvey. Participants had a median age of 32 years (IQR 19–44) and 36.6% were male. Both diagnostic type results, RDT and ECLIA, were available for 1,800 individuals.

The proportion of visited households that did not participate in the mortality survey due to refusal or absence of the household head was 11.7% in Lubumbashi and 21.6% in Abidjan. Non-participation of households was much higher in the serosurvey (61.2% Lubumbashi, 61.7% Abidjan). Additionally, individual refusals in households that participated in the serosurvey was high at 45% in both settings.

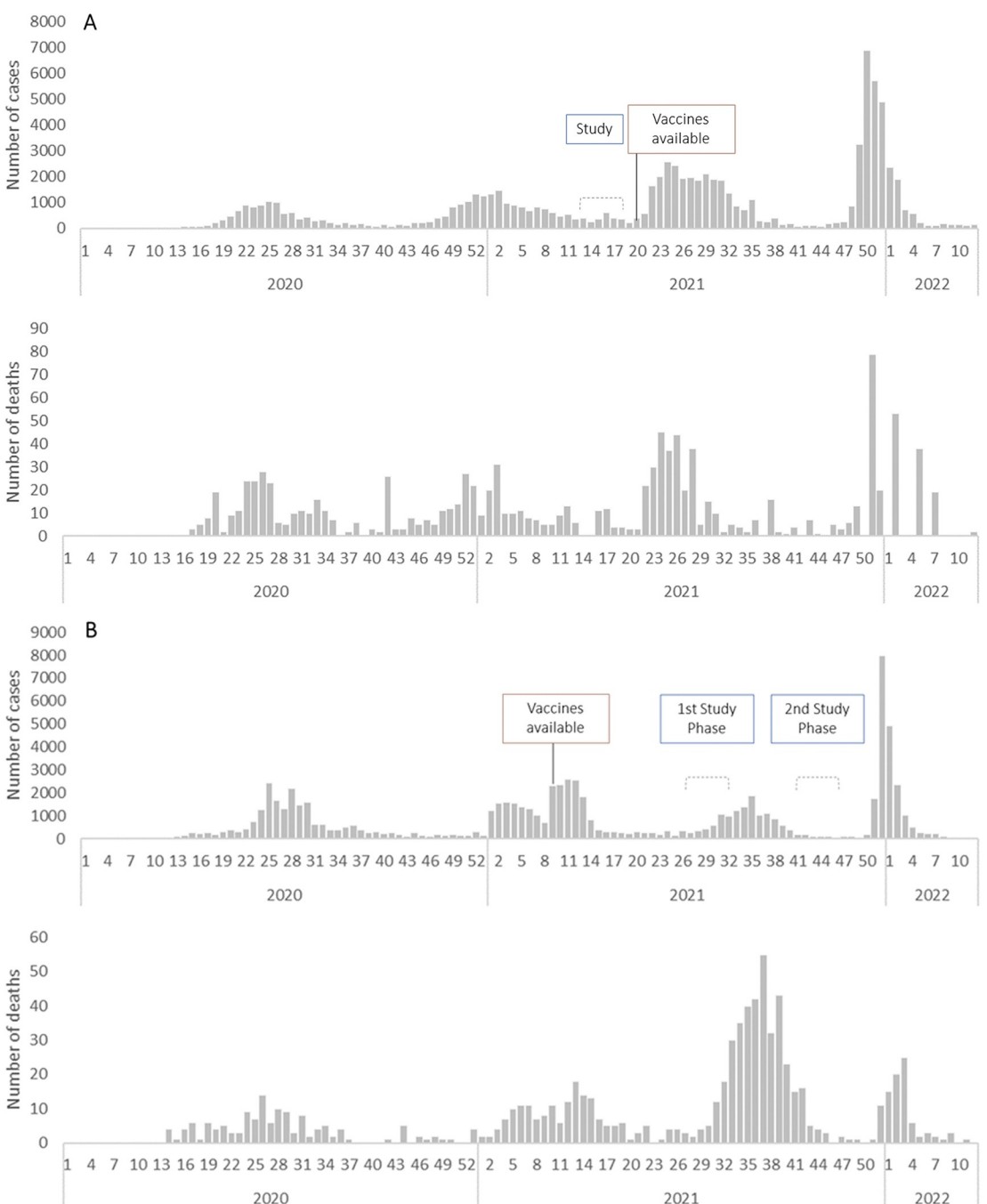

**Fig 1.** Timeline of Lubumbashi (**A**)\* and Abidjan (**B**) surveys and vaccine availability compared to the national progression of SARS-CoV-2 pandemic notified cases (upper curve) and deaths according to national surveillance systems [1] (lower curve). \**Note: 209 deaths reported week 16, 2020 in DRC are not represented due to the likely aggregation of data.*

## Mortality survey

In Lubumbashi, 150 deaths were reported by household heads during the entire recall period (1 January 2020–18 May 2021), including 17 deaths during the pre-pandemic and 133 deaths during the pandemic period (Fig 2A). In Abidjan, 83 deaths were reported by household heads during the recall period (1 January 2019–10 November 2021), including 29 deaths during the

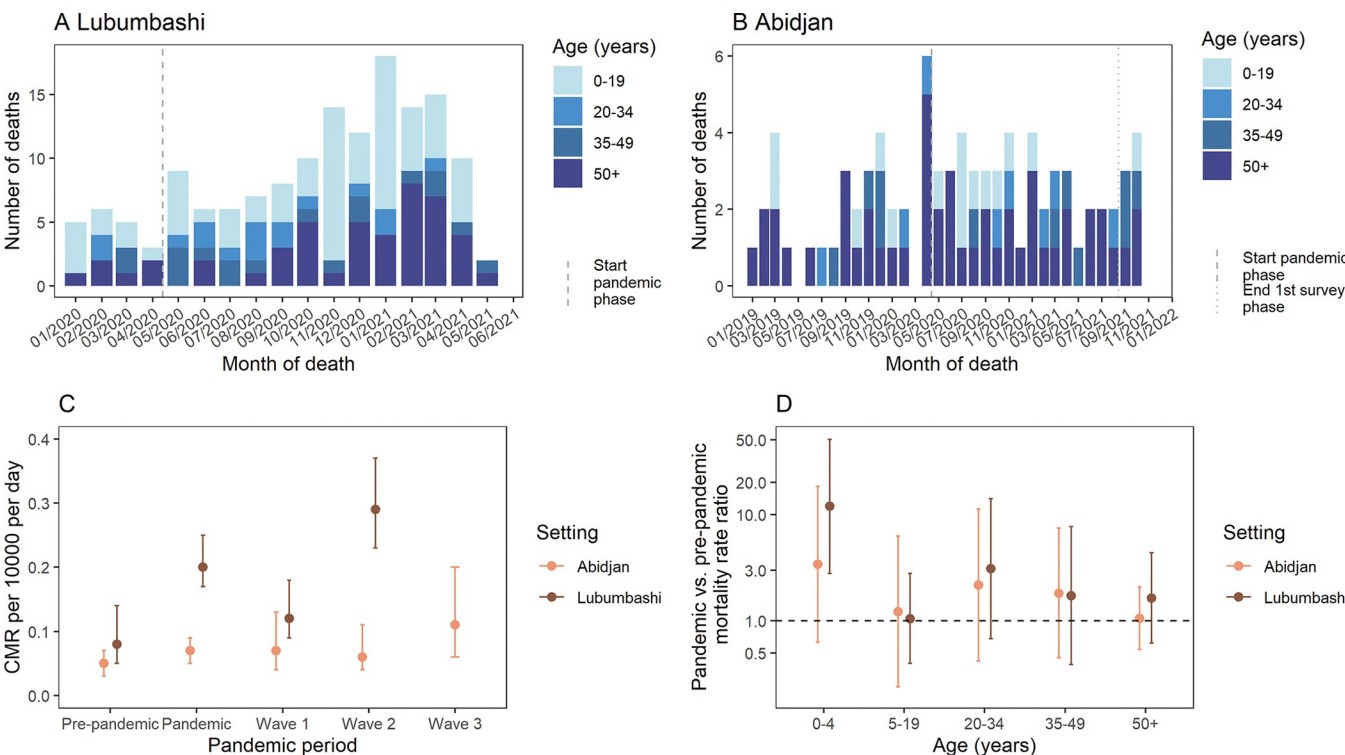

**Fig 2.** Number of reported deaths by age group over time in Lubumbashi (A) and Abidjan (B). CMR pre-pandemic and by pandemic periods (**C**). Pandemic vs. pre-pandemic mortality rate ratios by age group and setting (**D**).

pre-pandemic and 54 during the pandemic period (Fig 2B). The median age of reported deaths was 25 years (IQR 3–57) in Lubumbashi and 54 years (IQR 30–70) in Abidjan; 52.0% and 65.1% were male, respectively.

Fever/malaria was one of the main reported causes of death in both settings (Lubumbashi 34.0%, Abidjan 24.1%); respiratory diseases other than COVID-19 were indicated as the cause for 6.7% of deaths in Lubumbashi and 9.6% in Abidjan. Two deaths were attributed to COVID-19 in Lubumbashi, however occurred in the pre-pandemic period and are likely mis-classifications. In Abidjan no COVID-19 deaths were reported. The main reported causes of death did not change significantly between the pre-pandemic and pandemic period in either of the settings (Table A in S1 Text and Table B in S1 Text).

In Lubumbashi the overall CMR increased significantly from a pre-pandemic rate of 0.08 deaths per 10 000 persons per day (95%CI 0.05–0.14) to a pandemic rate of 0.20 deaths per 10 000 persons per day (95%CI 0.17–0.25) (Rate ratio [RR] = 2.5 [95%CI 1.4–4.3]; p = 0.001) (Fig 2C). The increase was statistically significant only among under 5-year-olds, however not among the other age groups (Fig 2D, Table C in S1 Text). In Abidjan no significant increase in the CMR was observed overall (pre-pandemic 0.05 deaths per 10 000 persons per day [95%CI 0.03–0.07], pandemic 0.07 deaths per 10 000 persons per day [95%CI 0.05–0.09]; RR 1.5 [95% CI 0.9–2.5], p = 0.099) (Fig 2C) or by age group (Fig 2D, Table D in S1 Text). The increase in CMR varied by SARS-CoV2 waves. In Lubumbashi, compared to the baseline period, the increase was statistically significant during the second wave (CMR wave 2: 0.29 deaths per 10 000 persons per day [95%CI 0.23–0.37]; RR 3.5 [95%CI 2.0–6.1], p<0.001) but not during the first wave (CMR wave 1: 0.12 deaths per 10 000 persons per day [95%CI 0.09–0.18]; RR 1.5 [95%CI 0.8–2.9], p = 0.216) (Fig 2C, Table E in S1 Text). In Abidjan, a significant increase in

CMR was observed only during the 3rd wave (CMR wave 3: 0.11 deaths per 10 000 persons per day [95%CI 0.06–0.20], RR 2.4 [95%CI 1.1–5.1]; p = 0.024) (Fig 2C, Table F in S1 Text). We performed a sensitivity analysis to investigate if observed increases during the pandemic period in Lubumbashi and the 3rd wave in Abidjan may have been caused by seasonal effects. The results of the sensitivity analysis showed consistent results with the main analysis (Table G in S1 Text and Table H in S1 Text).

Geographical differences in CMR were observed in Lubumbashi, where the health zones of Kampemba/Tshamilemba were more affected by the increase than the health zone of Lubumbashi (Table I in S1 Text). In Abidjan, the pattern was homogeneous across the two health zones (Table J in S1 Text). The risk of reporting a death in a household was associated with crowded living conditions; after adjusting for household size, households with an average of more than 1–2 household members per room (Lubumbashi OR 5.6 [95%CI 2.4–13.2]; Abidjan OR 8.1 [95%CI 1.1–61.0]) and households with more than 2 household members per room (Lubumbashi OR 8.1 [95%CI 3.2–20.5]; Abidjan OR 7.6 [95%CI 1.0–58.2]) were more likely to report a death than households with ≤1 household members per room. Moreover, in Abidjan the risk of reporting a death was higher among households living in a house with common courtyard than living in an individual house (OR 2.1 [95%CI 1.2–3.8]) (Table K in S1 Text and Table L in S1 Text).

## Seroprevalence survey

In Lubumbashi, 320/2038 participants were found seropositive by RDT, resulting in a weighted seroprevalence of 15.7% (95%CI 13.6–18.1) after the 2nd wave. In Abidjan, 445/1471 of unvaccinated participants had a positive RDT result. The weighted seroprevalence among unvaccinated doubled from 17.4% (95%CI 13.3–22.1) during phase 1 (after the 2nd wave) to 38.8% (95%CI 43.1–43.7) during phase 2 of the survey (after the 3rd wave).

ELISA/ECLIA seroprevalence was 1.8–3.4 times higher than based on RDT, with a weighted seroprevalence of 43.2% (95%CI 40.0–46.4) in Lubumbashi and 72.9% (95%CI 67.3–78.0) and 82.2% (95%CI 78.0–85.9) in phase 1 and 2 in Abidjan, respectively. A detailed comparison of RDT and ELISA results is provided in the Supplementary material (Table M in S1 Text and Table N in S1 Text).

In Lubumbashi, seroprevalence was highest among 50+ year-olds (RDT: 26.6% [95%CI 19.5–34.6]; ELISA: 52.4% [95%CI 43.5–61.2]) and lowest among 20–34-year-olds (RDT: 13.1% [95%CI 9.6–17.2]; ELISA: 40.8% [95%CI 35.2–46.5]) (Fig 3). The age pattern followed a similar tendency in Abidjan based on RDT for both phases of the survey, however the pattern was slightly different based on ECLIA (Fig 3).

Geographic differences were found in Lubumbashi based on RDT results, where seroprevalence was higher in Kampemba/Tshamilemba health zones than in Lubumbashi health zone; no difference was however observed based on ELISA results (Table O in S1 Text). In Abidjan, after considering the phase of the survey, no significant differences were observed between strata based on RDT or ECLIA results. The direction of differences between strata however varied by survey phase (Table P in S1 Text).

In both settings, and for both types of tests, the risk of being seropositive was higher in households with at least one other seropositive member compared with households with no other seropositive household member (Lubumbashi: OR 3.1 [95%CI 1.5–6.4]; Abidjan: OR 3.3 [95%CI 2.53–4.21]). However, the risk of having a deceased household member was not higher in households with one or more seropositive individuals (Lubumbashi OR 0.9 [95%CI 0.4–2.4]; Abidjan: OR 1.08 [95%CI 0.3–3.87]).

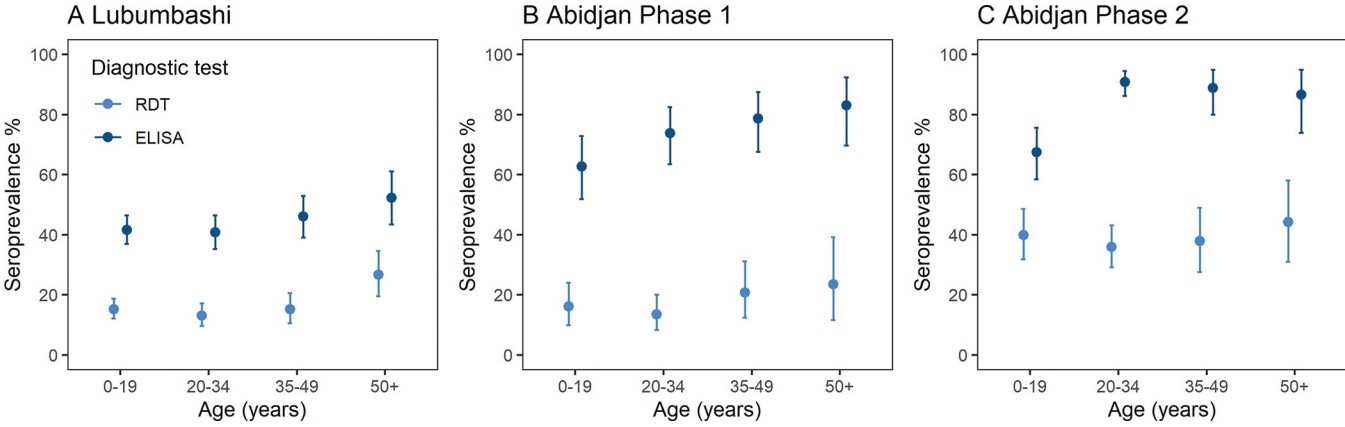

**Fig 3.** RDT and ELISA/ECLIA based seroprevalence by age group in Lubumbashi (**A**) and Abidjan for phase 1 (**B**) and phase 2 (**C**) of the survey.

The weighted proportion reporting COVID-19 related symptoms among RDT positive individuals was 22.4% (95%CI 16.2–29.5) in Lubumbashi and 71.6% (95%CI 65.5–77.2) in Abidjan. This proportion was similar among ELISA/ECLIA positive individuals (Lubumbashi 19.0% [95%CI 15.2–23.3]; Abidjan 71.8% [95%CI 65.7–77.4]).

We did not find significant positive associations of seropositivity and work related or social contact related exposures (Table Q in S1 Text and Table R in S1 Text).

## COVID-19 vaccination

In Lubumbashi, the survey took place before the roll-out of COVID-19 vaccination. In Abidjan, the weighted proportion of vaccinated individuals nearly doubled from 9.0% (95%CI 6.5–11.9) during the first phase of the survey to 16.8% (95%CI 13.8–20.2) during the second phase of the survey. Vaccination rates increased with age; 31.7% (95%CI%: 25.1–38.8) of those aged 50+ years reported having been vaccinated. While women (11.9% [95%CI%: 8.1–16.5]) had a higher vaccination rate than men (5.6% [95CI% 3.6–8.2]) in the first phase, no difference was observed in the second phase. Among the vaccinated individuals, 89.3% had a positive RDT and 97.7% a positive ECLIA result.

## Discussion

We present here two of the first COVID-19 studies that combine mortality and seroprevalence surveys, providing mortality estimates and seroprevalence of anti-SARS-CoV-2 antibodies in a representative sample of the general population after the second pandemic wave in Lubumbashi, before the spread of the Delta variant, and during and after the third pandemic wave in Abidjan during which the Delta variant circulated [13,14]. In Lubumbashi, the CMR overall doubled during the pandemic period compared to the pre-pandemic period, with a more pronounced increase during the second wave. Most affected were the <5 year-olds, suggesting that the increase in mortality may have, in addition to the pandemic, been driven by factors indirectly impacted by the pandemic such as delayed or decreased access to healthcare due to fear of contracting COVID-19, reduced health system capacity, disruption in service provision, transport restrictions or economic challenges [15,16]. In Abidjan, there was no significant increase in crude mortality rates between the pre-pandemic and pandemic phase. However, an increase was observed during the 3rd wave of COVID-19, consistent with the increase in deaths reported by the official COVID-19 surveillance system compared to previous waves.

Overall, the increases in mortality observed in both studies were low, especially when compared to the excess mortality observed in areas of Europe, Asia and the Americas [17–19].

Our results indicate that a large proportion, if not the majority, of the population has anti-SARS-CoV-2 antibodies in both settings. According to the laboratory-based results, more than 40% of the population was infected with SARS-CoV-2 during the first two waves in Lubumbashi and more than 80% during the first 3 waves in Abidjan. Our findings are similar to more recent seroprevalence studies in several African countries which indicate that, while heterogenous, the virus has circulated widely [3]. We found that seroprevalence increased with age, contrary to observations in European countries [20–22], but similar to other seroprevalence surveys in Africa [3]. The infection risk in the African context thus appears to be higher in the elderly population, who also have a higher risk of severe disease.

The estimated infection rates were tens (in Abidjan) to hundreds (Lubumbashi) times higher than confirmed cases in the surveillance system of both countries. Our seroprevalence results in Abidjan show similar spread in Yopougon as in Marcory, in contrast to surveillance system figures which indicated a much lower attack rate in Yopougon. In Abidjan, based on the ECLIA results from the 2nd phase and official notified cases, we estimated 1 in 31 infections were reported in Marcory and only 1 in 151 in Yopougon. In Lubumbashi, only 1 in 150 infections were reported in the Lubumbashi health zone and only 1 in 700 infections were reported in Kampemba and Tshamilemba health zones. These results are consistent with a previous seroprevalence study in Kinshasa (DRC), which estimated after the 1st wave that only 1 in 300 infections were reported [4].

Our studies had several limitations. First the refusal rate was high in both contexts, particularly for the seroprevalence survey, suggesting a risk of selection bias. The median age of participants was higher in the seroprevalence than mortality survey, suggesting that young individuals may have been more likely to refuse. The lengthy recall period may have biased the accuracy of the collected death information, particularly for deaths that occurred earlier in the period, and in some cases, deaths may have been missed or household members may have been omitted. COVID-related stigma may have resulted in underreporting of COVID-related deaths, as no death was directly attributed to COVID-19 in either study. Moreover, we were not able to adjust for potential remaining confounding such as seasonality of other circulating pathogens. A sensitivity analyses restricting the analysis to the same months during the pre-pandemic and the pandemic phase showed however consistent results.

The performance of ELISA/ECLIA and RDT are significantly different, with ELISA/ECLIA-based seroprevalence estimates 3 times higher than those estimated by RDT. Similar disparities have been observed in other seroprevalence studies [23,24]. Test performance varies with time since infection and severity of infection, and RDT sensitivity may be lower to detect infections that occurred early during the epidemic resulting in lower seroprevalence estimates [11,25]. One study observed 91.3% and 83.7% of COVID-19 positive healthcare workers tested positive for IgM and IgG, respectively, with Biosynex RDT one month after infection versus 51.8% IgM and 56.8% IgG at months 11–13 [26]. Another longitudinal study among public health workers found Euroimmun ELISA responses increased in the first month and then decreased for several weeks, with predicted proportion of sero-reverting of 59% after 12 months while Roche ECLIA responses increased, with no sero-reversions at week 24 and predicted sero-reverting of <2% at 12 months [27]. False-positive results due to cross-reactivity cannot be entirely excluded [28]. Additionally, symptoms experienced more than a year ago may be difficult to recall and are not specific, making accurate estimation of symptomatic infections not possible. To limit bias, participants were asked about their symptoms before they knew their RDT result.

In Abidjan, where COVID-19 vaccination campaigns had started at the time of survey, our testing methodology did not distinguish between antibodies developed following vaccination and/or infection and antibodies developed following infection only. Nevertheless, the sero-prevalence is comparably high among the non-vaccinated and total study population.

Lastly it is unclear to what extent SARS-CoV-2 cross-reactive antibodies provide protection against future SARS-CoV-2 infection or disease progression, particularly in the event of new variants. The results of this analysis predate the third wave of cases in Lubumbashi and subsequent waves in Abidjan.

Despite these limitations, our study provides key results estimating the extent of SARS-CoV-2 infections in two urban African contexts and additionally the impact of infections on mortality.

## Conclusion

Although circulation of SARS-CoV-2 seems to have been wide in both study settings, the public health impact of COVID-19 varied by setting and seems to have been overall low compared to Europe, Asia or the Americas. The results in Lubumbashi, with increases particularly among the youngest age group, suggest indirect impacts of COVID and the pandemic on population health. The seroprevalence results confirmed substantial underdetection of cases through the national surveillance systems and demonstrated an increased risk of infection among the oldest age group, who are also at risk of more severe disease progression. Lastly, due to the high overall spread of the virus, our results suggest that targeted vaccination campaigns are appropriate to protect higher risk populations.

## Supporting information

**S1 Text. Supplementary information and supplementary analyses.**
(DOCX)

**S2 Text. GENERIC QUESTIONNAIRES.**
(DOCX)

## Acknowledgments

We thank the survey participants and the field study teams for their time and participation. We further thank the community leaders of the health zones Lubumbashi, Kampemba, and Tshamilemba and the communes of Marcory and Yopougon, as well as the Ministry of Health in the DRC and in Côte d'Ivoire for their approval and support during the implementation of the surveys.

## Author Contributions

**Conceptualization:** Erica Simons, Birgit Nikolay, Alessandro Pini, Klaudia Porten, Francisco Luquero, Etienne Gignoux.

**Formal analysis:** Erica Simons, Birgit Nikolay.

**Investigation:** Erica Simons, Pascal Ouedraogo, Carlos Tiemeni, Kaouther Chamman, Mireille Dosso, Moussa Doumbia, Viviane Kouakou Akissi, Jacques Muzinga, Lou Penali, Halidou Salou, Daouda Sevede.

**Methodology:** Birgit Nikolay, Etienne Gignoux.

**Project administration:** Pascal Ouedraogo, Estelle Pasquier, Ismael Adjaho, Colette Badjo, Mariam Diomandé, Mireille Dosso, Yves Asuni Izia, Hugues Kakompe, Anne Marie Katsomya, Vicky Kij, Christopher Mambula, Basile Ngoy.

**Supervision:** Erica Simons, Birgit Nikolay, Estelle Pasquier.

**Visualization:** Erica Simons, Birgit Nikolay.

**Writing – original draft:** Erica Simons, Birgit Nikolay, Estelle Pasquier, Etienne Gignoux.

**Writing – review & editing:** Pascal Ouedraogo, Ismael Adjaho, Colette Badjo, Mariam Diomandé, Mireille Dosso, Moussa Doumbia, Yves Asuni Izia, Hugues Kakompe, Anne Marie Katsomya, Vicky Kij, Viviane Kouakou Akissi, Christopher Mambula, Placide Mbala-Kingebeni, Jacques Muzinga, Basile Ngoy, Lou Penali, Alessandro Pini, Klaudia Porten, Halidou Salou, Daouda Sevede, Francisco Luquero.

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
