## [Decision Letter · Decision Letter 0]

21 Feb 2023

PGPH-D-22-01970

Seroprevalence of SARS-CoV-2 antibodies and retrospective mortality in two African settings: Lubumbashi, Democratic Republic of the Congo and Abidjan, Côte d’Ivoire

Dear Dr. Nikolay,

Thank you for submitting your manuscript to PLOS Global Public Health. After careful consideration, we feel that it has merit but does not fully meet PLOS Global Public Health’s publication criteria as it currently stands. Therefore, we invite you to submit a revised version of the manuscript that addresses the points raised during the review process.

EDITOR: Please insert comments here and delete this placeholder text when finished. Be sure to:

Indicate which changes you require for acceptance versus which changes you recommendAddress any conflicts between the reviews so that it's clear which advice the authors should followProvide specific feedback from your evaluation of the manuscript

Please ensure that your decision is justified on PLOS Global Public Health’s publication criteria and not, for example, on novelty or perceived impact.

We look forward to receiving your revised manuscript.

Kind regards,

Julio Croda, Ph.D, M.D.

Academic Editor

Journal Requirements:

1. In the online submission form, you indicated that "The minimal dataset underlying the findings of this study is available on request, in accordance with the legal framework set forth by Médecins Sans Frontières (MSF) data sharing policy (Karunakara U, PLoS Med 2013). MSF is committed to share and disseminate health data from its programs and research in an open, timely, and transparent manner in order to promote health benefits for populations while respecting ethical and legal obligations towards patients, research participants, and their communities. The MSF data sharing policy ensures that data will be available upon request to interested researchers while addressing all security, legal, and ethical concerns. All readers may contact data.sharing@msf.org or Nouha TOUATI (nouha.touati@epicentre.msf.org) to request data.. All PLOS journals now require all data underlying the findings described in their manuscript to be freely available to other researchers, either 1. In a public repository, 2. Within the manuscript itself, or 3. Uploaded as supplementary information.

Additional Editor Comments (if provided):

Reviewers' comments:

Reviewer's Responses to Questions

**Comments to the Author**

1. Does this manuscript meet PLOS Global Public Health’s publication criteria? Is the manuscript technically sound, and do the data support the conclusions? The manuscript must describe methodologically and ethically rigorous research with conclusions that are appropriately drawn based on the data presented.

Reviewer #1: Yes

Reviewer #2: Partly

2. Has the statistical analysis been performed appropriately and rigorously?

Reviewer #1: Yes

Reviewer #2: Yes

3. Have the authors made all data underlying the findings in their manuscript fully available (please refer to the Data Availability Statement at the start of the manuscript PDF file)?

Reviewer #1: Yes

Reviewer #2: Yes

4. Is the manuscript presented in an intelligible fashion and written in standard English?

Reviewer #1: Yes

Reviewer #2: Yes

5. Review Comments to the Author

Reviewer #1: This paper presents the results of a retrospective mortality survey and cross-sectional survey in two locations in Africa – one in the DRC and one in Cote D’Ivoire. The finding of increased mortality during the pandemic among <5 year-olds is particularly striking and a timely reminder of the importance of maintaining access to healthcare in these settings. I think this is a worthwhile addition to the literature and I have minor comments only.

Minor comments:

Line 113: Why was January 1 2020 chosen for Lubumbashi and not January 1 2019? Could you perform a sensitivity analysis limiting the recall period to 1 January 2020 in Abidjan, to assess the possibility of bias due to the different recall periods and seasonality?

Line 117: I’m unclear on how the survey questions were worded here. Were household members asked how many deaths occurred in each of these periods of time (including the COVID-19 waves)? Or were they asked the exact date of death, and only asked about the periods if they could not recall the exact date of death? The wording “12 April 2020 was chosen as cut-off date between the pre-pandemic and pandemic period as it was the Easter Sunday, a date easy to recall for survey participants.” makes it seem like the former (i.e. exact date not ascertained), but this justification doesn’t make sense given the detail that is required in lines 118-120. Please clarify how the information on date of death was obtained.

Line 162 and 169: what does “matched” mean in this context?

Line 188-193: does “reported” here mean self-reported by household members?

Line 307-311: is there any data on the sensitivity of the RDT and EUROIMMUN over time since infection/symptom onset (i.e. the proportion of individuals who serorevert), that you could report here? Given the high se/sp reported in the methods for the RDT, and the results you have found, I suspect the RDT wanes quickly and therefore a positive RDT could be an indication of a “recent” infection. Therefore, it might be worth mentioning this interpretation, e.g. in your interpretation of the shape of the RDT seropositivity by age curve and the differences between Phase 1 and Phase 2 in Abidjan.

Reviewer #2: This article is important to PLOS Global Public Health to communicate new points of view and health information about differents locations of Africa, especially related to life conditions. However,

the submission declare several limitations, but their methodological and ethical designed research must be considerable too.

It is an important research for Africa epidemiology studies but some some gaps were observed:

1. The study would might be correlated with studies of genomic sequence that might be help to correlate with specific variants.

2. Lines 89 to 94 – “A detailed description of the study design and sampling methodology is provided in the supplementary information. The mortality questionnaire consisted of sections covering housing characteristics, demographic data, and information on deaths that occurred during the recall period. For households included in the seroprevalence survey, individual questionnaires were administered to each household member or their parent/guardian covering socio-demographic information, medical history, potential SARS-CoV-2 exposures, history of any possible COVID-related symptoms since 2020 and COVID-19 vaccination status.” Is it possible present any results of medical history, potential SARS-CoV-2 exposures, history of any possible COVID-related symptoms since 2020 from data collected?

3. Is it possible explain increased malaria deaths in Pandemic period? (Table S1 and S2).

4. Lines 292-294 - "In addition, our results show that the infection has spread as much in Yopougon as in Marcory, in contrast to surveillance system figures which indicated a much lower attack rate in Yopougon". Is it possible clarify more detailed this affirmation?

5.Is possible to add instruments of "mortality questionnaire (...) and of individual questionnaires(...)" lines 92-94?

6. PLOS authors have the option to publish the peer review history of their article (what does this mean?). If published, this will include your full peer review and any attached files.

**Do you want your identity to be public for this peer review?** For information about this choice, including consent withdrawal, please see our Privacy Policy.

Reviewer #1: No

Reviewer #2: No

---

## [Decision Letter · Decision Letter 1]

11 May 2023

Seroprevalence of SARS-CoV-2 antibodies and retrospective mortality in two African settings: Lubumbashi, Democratic Republic of the Congo and Abidjan, Côte d’Ivoire

PGPH-D-22-01970R1

Dear Dr Nikolay,

We are pleased to inform you that your manuscript 'Seroprevalence of SARS-CoV-2 antibodies and retrospective mortality in two African settings: Lubumbashi, Democratic Republic of the Congo and Abidjan, Côte d’Ivoire' has been provisionally accepted for publication in PLOS Global Public Health.

Best regards,

Julio Croda, Ph.D, M.D.

Academic Editor

Reviewer Comments (if any, and for reference):

Reviewer's Responses to Questions

**Comments to the Author**

1. If the authors have adequately addressed your comments raised in a previous round of review and you feel that this manuscript is now acceptable for publication, you may indicate that here to bypass the “Comments to the Author” section, enter your conflict of interest statement in the “Confidential to Editor” section, and submit your "Accept" recommendation.

Reviewer #1: All comments have been addressed

Reviewer #2: All comments have been addressed

2. Does this manuscript meet PLOS Global Public Health’s publication criteria? Is the manuscript technically sound, and do the data support the conclusions? The manuscript must describe methodologically and ethically rigorous research with conclusions that are appropriately drawn based on the data presented.

Reviewer #1: Yes

Reviewer #2: Yes

3. Has the statistical analysis been performed appropriately and rigorously?

Reviewer #1: Yes

Reviewer #2: Yes

4. Have the authors made all data underlying the findings in their manuscript fully available (please refer to the Data Availability Statement at the start of the manuscript PDF file)?

Reviewer #1: Yes

Reviewer #2: Yes

5. Is the manuscript presented in an intelligible fashion and written in standard English?

Reviewer #1: Yes

Reviewer #2: Yes

6. Review Comments to the Author

Reviewer #1: I thank the authors for addressing all my comments and congratulate them on an important paper

Reviewer #2: none

7. PLOS authors have the option to publish the peer review history of their article (what does this mean?). If published, this will include your full peer review and any attached files.

**Do you want your identity to be public for this peer review?** For information about this choice, including consent withdrawal, please see our Privacy Policy.

Reviewer #1: No

Reviewer #2: No

<quillbot-extension-portal></quillbot-extension-portal>